# Marine n-3 Polyunsaturated Fatty Acids and Bone Mineral Density in Kidney Transplant Recipients: A Randomized, Placebo-Controlled Trial

**DOI:** 10.3390/nu13072361

**Published:** 2021-07-10

**Authors:** Hanne Skou Jørgensen, Ivar Anders Eide, Trond Jenssen, Anders Åsberg, Jens Bollerslev, Kristin Godang, Anders Hartmann, Erik Berg Schmidt, My Svensson

**Affiliations:** 1Department of Kidney Diseases, Aarhus University Hospital, 8200 Aarhus, Denmark; 2Faculty of Health, Institute of Clinical Medicine, Aarhus University, 8000 Aarhus, Denmark; 3Department of Nephrology, Division of Internal Medicine, Akershus University Hospital, 1478 Lørenskog, Norway; Ivar.Anders.Eide@ahus.no (I.A.E.); m.h.s.svensson@medisin.uio.no (M.S.); 4Faculty of Medicine, Institute of Clinical Medicine, University of Oslo, 0372 Oslo, Norway; tjenssen@ous-hf.no (T.J.); anders.asberg@farmasi.uio.no (A.Å.); jens.bollerslev@medisin.uio.no (J.B.); kgodang@ous-hf.no (K.G.); anders.hartmann@medisin.uio.no (A.H.); 5Department of Transplantation Medicine, Oslo University Hospital Rikshospitalet, 0372 Oslo, Norway; 6Department of Pharmacy, University of Oslo, 0372 Oslo, Norway; 7Section of Specialized Endocrinology, Department of Endocrinology, Morbid Obesity and Preventive Medicine, Medical Clinic, Oslo University Hospital Rikshospitalet, 0372 Oslo, Norway; 8Department of Cardiology, Aalborg University Hospital, 9100 Aalborg, Denmark; ebs@rn.dk

**Keywords:** fatty acids, fish oil, bone density, chronic kidney disease-mineral and bone disorder, kidney transplantation, osteoporosis

## Abstract

Kidney transplant recipients are at high risk of progressive bone loss and low-energy fractures in the years following transplantation. Marine n-3 polyunsaturated fatty acids (n-3 PUFA) supplementation may have beneficial effects on bone strength. The Omega-3 fatty acids in Renal Transplantation (ORENTRA) trial was an investigator initiated, randomized, placebo-controlled trial investigating the effects of marine n-3 PUFA supplementation after kidney transplantation. Effects of supplementation on bone mineral density (BMD) and calcium metabolism were pre-defined secondary endpoints. Adult kidney transplant recipients (*n* = 132) were randomized to 2.6 g marine n-3 PUFA supplement or olive oil (control) from 8 to 52 weeks post-transplant. Dual energy X-ray absorptiometry was performed to assess changes in bone mineral density of hip, spine, and forearm, as well as trabecular bone score (TBS) of the lumbar spine. Student’s *t* test was used to assess between-group differences. There were no differences in ΔBMD between the two groups (intervention vs. control) at lumbar spine (−0.020 ± 0.08 vs. −0.007 ± 0.07 g/cm², *p* = 0.34), total hip (0.001 ± 0.03 vs. −0.005 ± 0.04, *p* = 0.38), or other skeletal sites in the intention-to-treat analyses. There was no difference in the change in TBS score (0.001 ± 0.096 vs. 0.009 ± 0.102, *p* = 0.62). Finally, no effect on biochemical parameters of mineral metabolism was seen. Results were similar when analyzed per protocol. In conclusion, we found no significant effect of 44 weeks of supplementation with 2.6 g of marine n-3 PUFA on BMD in kidney transplant recipients.

## 1. Introduction

Kidney transplant recipients are at high risk of fracture in the years following kidney transplantation [1,2,3]. Contributors include traditional risk factors such as age, gender [4], and ethnicity [5], pre-existing renal bone disease [6], ongoing disturbances of calcium- and phosphate-metabolism [7], pre-existing or new-onset diabetes mellitus, [8] and immunosuppressive therapy [5,9].

Marine n-3 polyunsaturated fatty acids (n-3 PUFA) are essential fatty acids with known anti-inflammatory properties [10]. n-3 PUFA may also have beneficial effects in bone [11], as in vivo studies have demonstrated inhibition of osteoclasts [12,13], and stimulation of osteoblasts [14], which could potentially translate into reduced bone resorption and increased bone mineralization. High levels of plasma marine n-3 PUFA was also associated with higher bone mineral density (BMD) [15,16] and a reduced risk of fractures [17]. Few studies have been published regarding the effect of n-3 PUFA supplementation on BMD and with inconsistent results reported [18,19]. Increased hip BMD was seen in elderly women treated for 18 months with 6 g of mixed oils, of which 420 mg were marine n-3 PUFA [18]. However, no effect was found of 12 months supplementation with 440 mg of marine fish oil on whole body BMD in pre- or post-menopausal women [19]. A positive association between plasma concentrations of marine n-3 PUFA and BMD was previously reported in kidney transplant recipients [20,21]; but no studies have yet investigated the effect of n-3 PUFA supplementation on bone metabolism in patients with chronic kidney disease.

The aim of this study was to investigate the effects of a moderate to high (≈2.6 g daily) supplement with marine n-3 PUFA on bone mass and mineral metabolism, as pre-defined secondary endpoints of a randomized, placebo-controlled trial in kidney transplant recipients.BMD determined at multiple sites, as well as trabecular bone score (TBS) of the lumbar spine, were included in the skeletal assessment.

## 2. Materials and Methods

### 2.1. Study Design and Cohort

This was a secondary endpoint analysis of the Omega-3 fatty acids in Renal Transplantation (ORENTRA) study, a randomized, double-blinded trial, conducted at the national transplant center at Oslo University Hospital Rikshospitalet in Norway between 2013 and 2015. A detailed description of inclusion and exclusion criteria has been published previously [22]. In brief, adult kidney transplant recipients with stable kidney function (>30 mL/min/1.73 m²) providing written, informed consent were included in the early post-transplant phase. Exclusion criteria were allergies to seafood or fish oil or kidney donor age >75 years. Participants (n = 132) were randomly assigned to receive either ≈2.6 g n-3 PUFA supplements (460 mg/g Eicosapentaenoic acid (EPA) + 380 mg/g Docosahexaenoic acid (DHA) (Omacor®, Pronova Biopharma, Oslo, Norway)), or extra virgin olive oil, given as 1 capsule of 1 g, three times daily. Treatment was initiated eight weeks post-transplant, and trial duration was 44 weeks. Thirty patients were excluded or withdrew from the study due to (treatment vs. control group): screening failure (1 vs. 1), gastrointestinal discomfort (9 vs. 8), serious adverse event (4 vs. 5), or dropout (2 vs. 0).

### 2.2. Immunosuppressive Protocol

Patients received induction therapy with basiliximab, followed by maintenance immunosuppressive therapy with prednisolone, mycophenolate, and the calcineurin inhibitor tacrolimus. One dose of methylprednisolone was given at the time of transplantation, followed by prednisolone 20 mg daily (day 0 to 14) tapered gradually to 5 mg per day at 6 months post-transplant. Tacrolimus dosage was adjusted according to trough concentrations, with a target of 3 to 7 μg/L. Prophylactic treatment with trimethoprim-sulfamethoxazole was used for 6 months, and valganciclovir was given to cytomegalovirus seronegative recipients with seropositive donors.

### 2.3. Biochemical Analyses

Fasting blood samples were drawn at the baseline visit and the last follow-up visit. Plasma intact parathyroid hormone (iPTH), phosphate, and total and ionized calcium were measured by Roche Modular E170 until 2016, then by the Roche Cobas e602 platform (Roche Diagnostics, Basel, Switzerland). All analyses were performed by an accredited laboratory (Department of Medical Biochemistry, Oslo University Hospital Rikshospitalet, Norway) according to standardized method protocols. Estimated glomerular filtration rate (eGFR) was calculated by the Chronic Kidney Disease Epidemiology Collaboration (CKD-EPI) equation, adjusting for age, gender, ethnicity, height, and weight.

Samples for plasma n-3 PUFA measurements were centrifuged, frozen, and stored at 80 °C at the Laboratory of Renal Physiology Biobank at Oslo University Hospital, Rikshospitalet, Norway. Aliquots of bio-banked plasma were sent to The Lipid Research Center, Aalborg University Hospital, Denmark, for fatty acid (FA) analysis, which was performed in four steps: (1) extraction of total lipids, performed by a modified Folch method [23]; (2) isolation of the phospholipid fraction as described by Burdge [24]; (3) transmethylation of phospholipid FAs, and finally; (4) quantification of FAs using a Varian 3900 gas chromatograph with a CP-8400 autosampler, a flame ionization detector and a CP-Sil 88 60 m × 0.25 mm capillary column (Varian, Middleburg, The Netherlands). FAs were identified from their relative retention time, and quantitated as the weight percent of total fatty acids (wt%). Total marine n-3 PUFA level was defined as the sum of EPA and DHA. Coefficients of variation (CV) for the analyses of EPA and DHA were 1.1% and 1.8%, respectively.

### 2.4. Bone Density

A dual energy X-ray absorptiometry (DXA) scan (GE Medical Systems, Lunar Corp., Madison, WI, USA) was performed at baseline and end of study to determine BMD at whole body, lumbar spine, proximal femur (total hip and femoral neck), and the non-dominant forearm (proximal and ultra-distal radius). Bone density is reported as absolute BMD in g/cm², with the addition of T-, and Z-scores calculated from normative data provided by the manufacturer. The Lunar reference database has previously been validated for clinical use in this population [25]. Standard imaging and positioning protocols were used, and all scans and subsequent analyses were performed by Certified Densitometry Technologists (CDT, The International Society for Clinical Densitometry, Middletown, CT, USA). Quality assurance check was carried out twice weekly, using an aluminum spine phantom in water (Lunar 17810, GE Medical Systems, Madison, WI, USA). Short- and long-term CV were 0.8% and 1.4%, respectively. The TBS parameter was retrospectively extracted from the DXA images of lumbar spine L1–L4, by using the TBS iNsight software v2.1.2.0 (Medimaps Group SA, Geneva, Switzerland). TBS at L1–L4 has a short-term in vivo precision of 1.1% to 1.9%.

### 2.5. Ethics

The ORENTRA trial was approved by the Regional Committees for Medical and Health Research Ethics (identifier 2012/1419) and The Norwegian Medicines Agency, and was performed in accordance with Good Clinical Practice and the Declaration of Helsinki. The study was registered at ClinicalTrials.gov (identifier NCT01744067), and the European Union Drug Regulating Authorities Clinical Trials Database (identifier 2012-004992-37; 5 February 2013).

### 2.6. Statistics

Data are expressed as mean ± standard deviation (SD), median with interquartile range (IQR), or *n* (%) as appropriate. Endpoints were analyzed in both the intention-to-treat (ITT) and per-protocol (PP) populations. Normality was assessed by qq-plots and the Shapiro-Wilk normality test. Differences in Δ values between groups were evaluated by Student’s *t* test, and associations between continuous variables were evaluated by Spearman’s correlations. Skewed data were transformed to their natural logarithm to enable parametrical testing. For all analyses, a two-sided *p* value < 0.05 was considered statistically significant. Statistical analysis was performed using software package Stata/IC 13.1 for Windows (StataCorp LP, College Station, TX, USA).

## 3. Results

Baseline demographic variables are presented in Table 1. Two-thirds of patients received dialysis therapy prior to transplantation, either in the form of chronic intermittent hemodialysis (*n* = 60, median duration 17 months, range 1 to 61), or peritoneal dialysis (*n* = 31, median duration of 10 months, range 1 to 34). Delayed graft function was seen in 11.4%.

Baseline plasma levels of n-3 PUFA were positively correlated with age (Spearman’s rho = 0.42, *p* < 0.001), and negatively correlated with eGFR (rho = −0.18, *p* = 0.04). There were no associations with gender, body mass index, diabetes mellitus prior to transplantation, new onset diabetes mellitus, or biochemical measures of calcium metabolism. Baseline n-3 PUFA content in plasma phospholipids was positively correlated with *Z*-scores of L1-L4 lumbar spine (rho = 0.26, *p* = 0.003), total hip (rho = 0.24, *p* = 0.007) and femoral neck (rho = 0.19, *p* = 0.03), but not with *Z*-scores of whole body (rho = 0.08, *p* = 0.36), proximal radius (rho = 0.11, *p* = 0.23), or distal radius (rho = 0.14, *p* = 0.13). There was also no significant correlation between n-3 PUFA and the L1–L4 TBS (rho = −0.05, *p* = 0.57).

Table 2 shows changes in plasma marine n-3 PUFA levels in relation to biochemical markers of calcium metabolism, and BMD after 44 weeks. Supplementation resulted in a significant increase in plasma n-3 PUFA content, from 6.04 to 10.7 wt% *(p* < 0.001) in the intervention group, with no change in the control group (6.02 to 6.27 wt%, *p* = 0.43). Plasma levels of iPTH, calcium, and phosphate were unaffected by marine n-3 PUFA supplementation.

There were no differences in ΔBMD at the whole body, lumbar spine, proximal femur, or forearm between the two groups (Figure 1), and the same was true for Δ-values of the lumbar spine TBS score.

Further, we found no significant correlations between the increase in plasma level of marine n-3 PUFA and ΔBMD, at the lumbar spine, the total hip, or the distal forearm (Figure 2). A significant inverse correlation was found between baseline plasma levels of marine n-3 PUFA and change in lumbar spine BMD (*r* = −0.25, *p* = 0.006; Appendix A). However, no significant between-group differences were seen when restricting analyses to patients with below median levels of marine n-3 PUFA at baseline (Appendix A). ΔBMD of other skeletal sites measured were not associated with baseline n-3 PUFA levels.

## 4. Discussion

We found no effect of a 2.6 g daily combined EPA + DHA supplement on BMD or biochemical markers of calcium metabolism. These were pre-specified, secondary endpoints of the ORENTRA trial, and to our knowledge this is the first randomized, placebo-controlled trial to consider the effects of marine n-3 PUFA on bone disease and mineral metabolism after kidney transplantation.

A priori there were indications that marine n-3 PUFA might positively affect bone strength and reduce fracture risk in kidney transplant recipients. Direct effects of EPA and DHA on bone cell maturation, function, and apoptosis, favoring bone formation over resorption, have been demonstrated in several in vitro studies [12,13,14]. Positive effects of n-3 PUFA supplementation on bone mass, BMD, and bone strength have also been consistent findings in animal models [26]. Further, an indirect effect of n-3 PUFA on bone through increased calcium absorption in the intestines has been reported in experimental models [27,28]. In our previous observational study, a positive correlation was found between plasma n-3 PUFA and total calcium levels in kidney transplant recipient, which might support a similar mechanism in humans [21]. However, in our present interventional study, we were unable to demonstrate an effect of marine n-3 PUFA supplementation on biochemical measures of mineral metabolism. We also found no effect of this intervention on the TBS score, a gray-scale textural analysis of DXA images of the lumbar spine. The TBS is an index of trabecular microstructure, with the potential to deliver information on bone quality which is not readily captured by BMD [29].

The possibility of a threshold effect of n-3 PUFA on BMD has been proposed as an explanation for inconsistent results in observational studies. Two large cohorts from regions of low fish intake, the Women’s Health Initiative (WHI) [30] and the UK Framingham Osteoporosis study (FOS) [31], reported no associations between measured levels of n-3 PUFA and BMD. In contrast, positive correlations were found between n-3 PUFA levels and peak bone mass in young Swedish men [15], and total hip T-scores in Korean women [16]. Similarly, we found positive correlations between marine n-3 PUFA levels and baseline Z-scores in our study. It is possible that a significant proportion of our patients were already above a threshold of marine n-3 PUFA optimal for bone health, and that further supplementation had little potential to provide additional benefit.

Two previous studies investigated the association between n-3 PUFA and BMD after kidney transplantation. Baggio et al. reported a positive association between change in plasma n-3 PUFA level and ΔBMD over a two-year period [20], in a small study of 19 kidney transplant recipients. We could not confirm this association in our present study, though we did explore correlations between increase in plasma n-3 PUFA and change in BMD. In our previous study, we reported a positive association between plasma values of n-3 PUFA and BMD Z-scores of spine and hip in a large cohort of kidney transplant recipients at 6–8 weeks post-transplant [21]. This association was robust despite adjustment for multiple potential confounders; however, regression coefficients indicated a very modest effect-size. Our present study sample size may therefore not have been large enough, and the time frame of 44 weeks may also have been too short, to detect modest changes in BMD.

A recent meta-analysis summarized interventional trials investigating the effect of n-3, n-6 and a mixture of PUFAs on musculoskeletal health. The authors concluded that n-3 PUFA supplementation may result in a small (2.6%) increase in lumbar spine BMD, based on the combined results of five studies totaling 463 participants. No significant effect was seen on femoral neck BMD [32]. However, the overall level of evidence of the included studies in this meta-analysis was considered of low to very low quality, and the authors’ note that there was considerable heterogeneity both in the populations studied, and the doses of n-3 PUFA used. The only trial with a low risk of bias was on an Australian cohort of 202 patients with knee osteoarthritis randomized to high (4.5 g EPA + DHA) or low (0.45 g) dose n-3 PUFA supplementation. In this study, no effect of high dose n-3 PUFA was found on lumbar spine or femoral neck BMD after two years of treatment [33].

Study design and the complete follow-up of the cohort are strengths of this study. When considering BMD as an endpoint, the follow-up time was rather short, particularly as recent studies indicate highly variable changes in BMD by DXA during the first year post-transplant [34,35]. The clinical value of BMD monitoring by DXA-scans within a time-period of 1–3 years is debated [36,37], but on the other hand, recent international guidelines do suggest repeated DXA 1 year after initiation of therapy [38]. An even longer time-interval may be necessary to detect changes in the lumbar spine TBS, as the least significant change of this parameter is reported to be higher than that of BMD using the same DXA equipment [39]. Norwegians are known for a high-intake of fish, which may have diluted our intervention and potentially masked a true association. Thus, results may not be applicable to other populations with lower intakes of seafood. The supplementation protocol did, however, succeed in achieving a sizeable increase in plasma n-3 PUFA content, with a significant between-group difference after 44 weeks. Finally, the size of our cohort may not have been large enough to detect a modest effect of marine n-3 PUFA supplementation on BMD.

## 5. Conclusions

Our findings do not support recommending a supplement of marine n-3 PUFA to benefit bone health during the first year after kidney transplantation.

## Figures and Tables

**Figure 1 nutrients-13-02361-f001:**
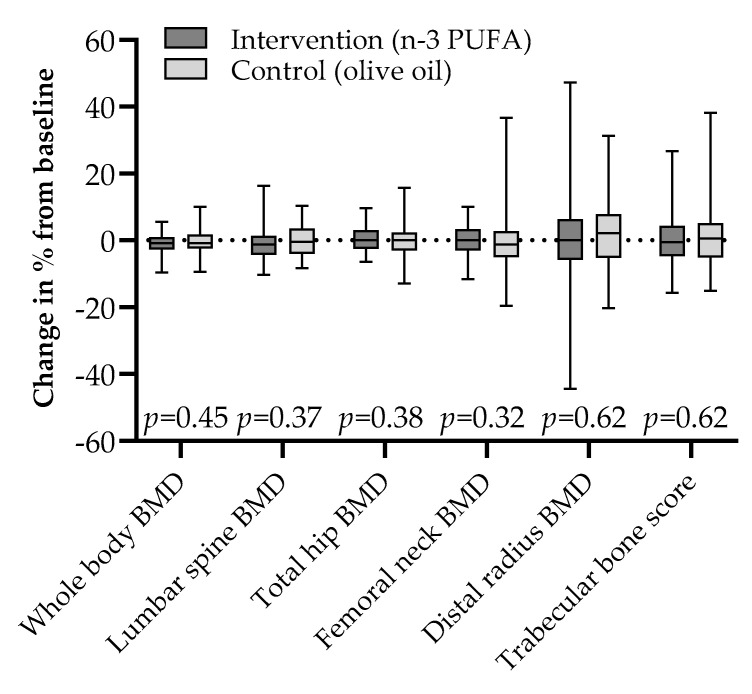
Changes in bone mineral density and trabecular bone score by intention to treat after 44 weeks of 2.6 g daily supplementation with marine n-3 polyunsaturated fatty acids or olive oil in adult kidney transplant recipients; median (line) with interquartile range (box) and full range (whiskers).

**Figure 2 nutrients-13-02361-f002:**
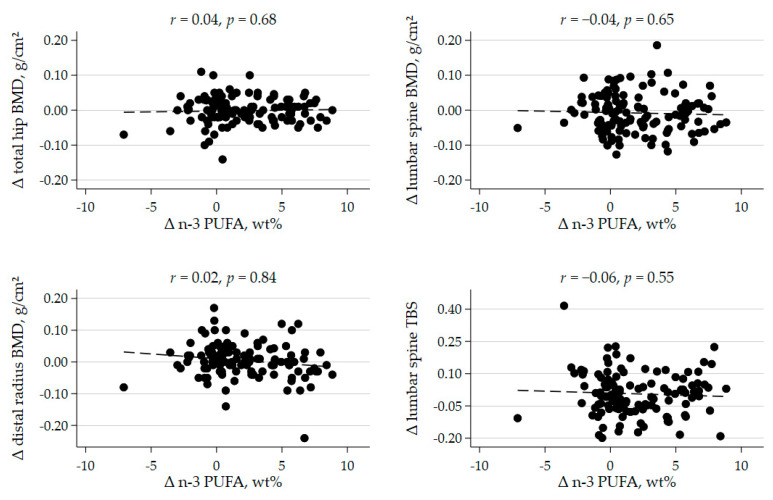
Scatterplots of the correlation between change in plasma marine n-3 polyunsaturated fatty acid (n-3 PUFA) concentration and change in bone mineral density (BMD) and trabecular bone score (TBS) after 44 weeks, *r* = Pearson’s correlation coefficient, with corresponding *p* value.

**Table 1 nutrients-13-02361-t001:** Baseline data of participating adult kidney transplant recipients.

Characteristic	All (*n* = 132)	Marine n-3 PUFA (*n* = 66)	Control (Olive Oil) (*n* = 66)
Age, years	53 ± 14	53± 14	54 ± 14
Women, %	34 (26%)	19 (29%)	15 (23%)
Caucasian, %	122 (92%)	60 (91%)	62 (94%)
Weight, kg	80 ± 15	79 ± 15	81 ± 15
Body mass index, kg/m²	26.0 ± 3.9	25.7 ± 3.8	26.2 ± 4.0
University degree, *n* (%)	49 (37%)	18 (30%)	29 (44%)
Exercize ≥ 2 per week	56 (42%)	30 (45%)	26 (39%)
Active smoker, %	23 (17%)	12 (18%)	11 (17%)
Daily use of fish oil supplement, %	19 (14%)	8 (12%)	11 (17%)
Dialysis pre-transplant, %	90 (68%)	45 (68%)	45 (68%)
Living donor, %	32 (24%)	14 (21%)	18 (27%)
HLA mismatches			
None or 1	21 (16%)	9 (14%)	12 (18%)
2 or 3	68 (51%)	34 (51%)	34 (51%)
≥4	43 (33%)	23 (35%)	20 (30%)
eGFR, mL/min/1.73m²	69 ± 21	71 ± 21	67 ± 22
Total n-3 PUFA, wt%	6.0 (4.7, 7.3)	6.0 (4.6, 7.6)	6.0 (4.8, 7.2)
Intact parathyroid hormone, ρmol/L	11.1 (9.1, 16.5)	10.9 (9.3, 16.4)	11.5 (9.1, 16.7)
Calcium ion, mmol/L	1.29 ± 0.07	1.30 ± 0.06	1.28 ± 0.06
Phosphate, mmol/L	0.84 ± 0.22	0.85 ± 0.23	0.86 ± 0.23
Whole body T-score	−0.38 ± 1.27	−0.40 ± 1.10	−0.36 ± 1.45
Lumbar spine T-score	−0.74 ± 1.44	−0.73 ± 1.34	−0.75 ± 1.55
Total hip T-score	−1.42 ± 0.99	−1.38 ± 0.86	−1.46 ± 1.12
Femoral neck T-score	−1.62 ± 1.05	−1.66 ± 0.86	−1.59 ± 1.23
Distal radius T-score	−0.74 ± 1.93	−0.92 ± 1.67	−0.55 ± 2.17
Lumbar spine TBS T-score	−2.35 ± 1.37	−2.35 ± 1.37	−2.34 ± 1.38

Data are mean ± standard deviation (SD), median with interquartile range (IQR), or *n* (%). Abbr.: HLA=human leukocyte antigen (-A, -B, and -DR), eGFR=estimated glomerular filtration rate by the CKD-EPI equation, PUFA=polyunsaturated fatty acids, TBS=trabecular bone score.

**Table 2 nutrients-13-02361-t002:** Effects of marine n-3 polyunsaturated fatty acid supplementation on bone mineral density in adult kidney transplantation recipients, by intention to treat and per protocol analyses.

Outcome Variable	*n*	Marine n-3 PUFA	*n*	Control (Olive Oil)	*p*
Fatty acids					
DHA + EPA, wt%					
ITT	61	4.00 ± 2.68	65	0.22 ± 2.22	<0.001
PP	50	4.44 ± 2.50	52	0.16 ± 2.28	<0.001
Bone density					
Whole body, g/cm²					
ITT	61	−0.009 ± 0.032	61	−0.001 ± 0.080	0.45
PP	49	−0.008 ± 0.033	50	−0.009 ± 0.046	0.87
Lumbar spine, g/cm²					
ITT	61	−0.014 ± 0.059	64	−0.005 ± 0.052	0.37
PP	49	−0.011 ± 0.062	51	−0.003 ± 0.053	0.45
Total hip, g/cm²					
ITT	61	0.001 ± 0.031	62	−0.005 ± 0.043	0.38
PP	49	0.000 ± 0.032	51	−0.004 ± 0.042	0.53
Femoral neck, g/cm²					
ITT	60	−0.000 ± 0.036	62	−0.009 ± 0.061	0.32
PP	48	0.001 ± 0.037	51	−0.009 ± 0.062	0.35
Distal radius, g/cm²					
ITT	61	0.002 ± 0.058	62	0.007 ± 0.049	0.62
PP	49	−0.001 ± 0.057	51	0.004 ± 0.050	0.63
Lumbar spine TBS					
ITT	61	0.001 ± 0.096	64	0.009 ± 0.102	0.62
PP	49	−0.004 ± 0.100	51	0.014 ± 0.104	0.38
Biochemistry					
Intact PTH, ρmol/L					
ITT	61	−1.2 ± 5.3	66	−2.0 ± 5.9	0.44
PP	49	−1.1 ± 4.5	52	−1.9 ± 5.7	0.41
25-OH Vit D, nmol/L					
ITT	58	10.5 ± 3.6	58	6.74 ± 3.2	0.44
PP	53	8.06 ± 3.7	50	6.16 ± 3.5	0.71
Calcium ion, mmol/L					
ITT	61	−0.01 ± 0.05	65	0.00 ± 0.06	0.47
PP	50	−0.00 ± 0.05	51	0.00 ± 0.05	0.55
Phosphate, mmol/L					
ITT	62	−0.15 ± 0.24	66	0.11 ± 0.23	0.45
PP	50	0.13 ± 0.24	52	0.13 ± 0.20	0.96

Data are mean ± SD of Δ-values after 44 weeks of intervention, with corresponding *p* values by Student’s *t* test for between-group differences, ITT = Intention to treat, PP = per protocol.

## Data Availability

The data presented in this study are available upon reasonable request from the corresponding author. The data are not publicly available due to privacy of participants.

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
