# Peer review of "Marine n-3 Polyunsaturated Fatty Acids and Bone Mineral Density in Kidney Transplant Recipients: A Randomized, Placebo-Controlled Trial"

_nutrients, 2021, doi:10.3390/nu13072361_

Round 1
Reviewer 1 Report
Jørgensen et al report on a randomized placebo-controlled intervention trial comparing the effects of omega-3 fatty acids to a placebo on bone mineral density in kidney transplant recipients. Basis was the “Orentra” trial, investigating the primary endpoint * in 152 kidney transplant recipients, where * received 2.6 g / day Omacor (containing 1.2 g EPA plus 1 g DHA per day), while * received an olive-oil placebo in weeks 8 to 52 post-transplant. A total of 30 participants did not conclude the trial, and the data reported are based on some 60 participants on verum and some 62 participants on placebo in the intention to treat analysis, and on some 50 participants on verum and some 51 participants on placebo in the per protocol analysis. Plasma fatty acid composition was analyzed in Aalborg with a method established in this lab. Here, the pre-defined secondary endpoints bone mineral density and calcium metabolism are reported. Bone mineral density was assessed with DEXA of hip, spine and forearm, and a trabecular bone score of the lumbar spine was also determined. No differences were found in these measurements between groups, both in intention to treat and clinical efficacy analysis. Changes in bone mineral density did not correlate with changes in plasma omega-3 fatty acids. The authors concluded “we found no significant effect of supplementation with 2.6 g of marine n-3 PUFA on BMD in kidney transplant recipients.”
Introduction is tight and concise, and accurately develops the research question. Materials and Methods are clearly described. Results are clearly and understandably presented. Discussion is short, but contains introductory elements.
Major Points
The trial was conducted in a Norwegian population, notorious for high intake background intake of EPA and DHA, and, consequently, for high levels of EPA and DHA in blood. The authors refrained from using a standardized fatty acid analysis of, e.g., red cell fatty acids, which could have demonstrated whether the trial was conducted in a population with high baseline levels or not. As is, it is unclear, whether the participants’ baseline omega-3 levels left any room for improvement, or whether participants’ baseline levels were so high that they could not be improved. This makes it unclear, whether the population recruited was adequate for the trial, i.e. whether the parameters measured could be improved by EPA and DHA.
No case estimate is reported for analyses presented. In light of the neutral results, by definition, it remains unclear, whether the trial was too small to detect an effect, or whether the effect was too small to be detected.
Taken together, it is unclear, whether the trial was conducted in a adequate population, and whether it was adequately sized. Each of both aspects largely invalidates data and conclusions.
Author Response
We thank the reviewers for their insightful comments on our manuscript. Please find our point-to-point replies below.
Reviewer #1
Jørgensen et al report on a randomized placebo-controlled intervention trial comparing the effects of omega-3 fatty acids to a placebo on bone mineral density in kidney transplant recipients. Basis was the “Orentra” trial, investigating the primary endpoint * in 152 kidney transplant recipients, where * received 2.6 g / day Omacor (containing 1.2 g EPA plus 1 g DHA per day), while * received an olive-oil placebo in weeks 8 to 52 post-transplant. A total of 30 participants did not conclude the trial, and the data reported are based on some 60 participants on verum and some 62 participants on placebo in the intention to treat analysis, and on some 50 participants on verum and some 51 participants on placebo in the per protocol analysis. Plasma fatty acid composition was analyzed in Aalborg with a method established in this lab. Here, the pre-defined secondary endpoints bone mineral density and calcium metabolism are reported. Bone mineral density was assessed with DEXA of hip, spine and forearm, and a trabecular bone score of the lumbar spine was also determined. No differences were found in these measurements between groups, both in intention to treat and clinical efficacy analysis. Changes in bone mineral density did not correlate with changes in plasma omega-3 fatty acids. The authors concluded “we found no significant effect of supplementation with 2.6 g of marine n-3 PUFA on BMD in kidney transplant recipients.”
Introduction is tight and concise, and accurately develops the research question. Materials and Methods are clearly described. Results are clearly and understandably presented. Discussion is short, but contains introductory elements.
Major Points
The trial was conducted in a Norwegian population, notorious for high intake background intake of EPA and DHA, and, consequently, for high levels of EPA and DHA in blood. The authors refrained from using a standardized fatty acid analysis of, e.g., red cell fatty acids, which could have demonstrated whether the trial was conducted in a population with high baseline levels or not. As is, it is unclear, whether the participants’ baseline omega-3 levels left any room for improvement, or whether participants’ baseline levels were so high that they could not be improved. This makes it unclear, whether the population recruited was adequate for the trial, i.e. whether the parameters measured could be improved by EPA and DHA.
Reply: We fully agree that the high intake of fish in the Norwegian population must be taken into account when interpreting our findings. While we acknowledge that erythrocyte content indeed provide a better indicator for long-term intake of marine n-3 PUFA, we assume that the baseline plasma n-3 PUFA measurement is representative of the usual diet of the participants. This is supported by dietary reports from this cohort, demonstrating that the intake of marine n-3 PUFA was stable across the study period (Chan Plos One 2020 17;15(12):e0244089). The baseline plasma n-3 PUFA content was median 6.0 wt%, i.e. half the cohort were within the target range of 6-8 wt%, suggested as optimal for cardiovascular health. Considering the possibility of a threshold effect of n-3 PUFA, this relatively high baseline value of n-3 PUFA is indeed an important limitation to consider; and is included in our discussion (page 7, lines 223-225 and 257-260).
No case estimate is reported for analyses presented. In light of the neutral results, by definition, it remains unclear, whether the trial was too small to detect an effect, or whether the effect was too small to be detected.
Reply: Bone health was a (pre-specified) secondary end-point of this trial. The primary end-point was measured glomerular filtration rate at 1 year post-transplant, and the power estimation for the trial was thus based on the expected effect on mGFR, with consideration of a drop-out rate of 20% (Eide, Am J Transplant 2019;19(3):790). Considering the highly variable evolution of BMD in the first year post-transplant (Evenepoel Kidney Int 2019;95:1461), we consider it likely that the cohort size was too small to detect a presumably modest effect of n-3 PUFA on bone health. We refer to the discussion (page 7, lines 233-237), and the concluding remark (page 8, lines 263-264), and hope that this important limitation is transparently discussed.
Taken together, it is unclear, whether the trial was conducted in a adequate population, and whether it was adequately sized. Each of both aspects largely invalidates data and conclusions.
Reply: We fully agree that these analyses, due to limited sample size and study duration, cannot provide firm conclusions on the effects of n-3 PUFA on bone health. We hope that this comes across in our discussion and limitation. Our conclusion is quite simply, that based on these findings, we cannot recommend supplementation with n-3 PUFA to benefit bone health in the first post-transplant year. We hope, however, that our study can be considered a valuable addition to the (limited) literature on this subject.
Reviewer 2 Report
I read with pleasure the work of Hanne Skou Jørgensen et al that investigates a potential protective role of marine n-3 PUFA oral supplementation on bone mass in a population of kidney transplanted patients.
This analysis is a part of the pre-defined secondary endpoints of ORENTRA trial (a randomized placebo-controlled trial designed to evaluate the effects of n-3 PUFA supplementation on renal function and cardiovascular risk markers in kidney transplanted population).
This research shows an absence of protective effect after 44 weeks of n-3 PUFA daily supplementation on bone mass and bone metabolism.
To observe changes in BMD and TBS values requires longer time than 44 weeks as considered in this study. Furthermore, the authors did not consider to report occurrence of clinical events such as manifested and/or clinically silent fractures (for example by means of Xray scan of the lumbar and sacral dorsal spine) nor specific markers of bone metabolism such as bone isoenzyme of alkaline phosphatase or serum CTX test.
Therefore, the data presented by the authors in such a small sample, in a single population, does not allow to draw any conclusion on the n-3 PUFA supplementation effect to bone mass and mineral metabolism. To investigate this effect, an ad hoc randomized trial would be required.
Author Response
Reviewer #2
I read with pleasure the work of Hanne Skou Jørgensen et al that investigates a potential protective role of marine n-3 PUFA oral supplementation on bone mass in a population of kidney transplanted patients.
This analysis is a part of the pre-defined secondary endpoints of ORENTRA trial (a randomized placebo-controlled trial designed to evaluate the effects of n-3 PUFA supplementation on renal function and cardiovascular risk markers in kidney transplanted population).
This research shows an absence of protective effect after 44 weeks of n-3 PUFA daily supplementation on bone mass and bone metabolism.
To observe changes in BMD and TBS values requires longer time than 44 weeks as considered in this study. Furthermore, the authors did not consider to report occurrence of clinical events such as manifested and/or clinically silent fractures (for example by means of Xray scan of the lumbar and sacral dorsal spine) nor specific markers of bone metabolism such as bone isoenzyme of alkaline phosphatase or serum CTX test.
Reply: Thank you for your comments. We do consider the short time-frame a major limitation for this secondary outcome of bone health. For post-menopausal osteoporosis, an interval of 1-2 years is recommended for repeat DXA-scans. However, in the first year after kidney transplantation, bone loss is marked in subsets of patients, and well above the least significant change of DXA, which is estimated at 2% (Evenepoel Kidney Int 2019;95:1461). Unfortunately, we did not have the opportunity to evaluate clinical or silent fractures in this cohort, and considering the cohort size, we do not believe we would have achieved sufficient events for a meaningful analysis. We also do agree that evaluation of biochemical bone turnover markers would have been of great interest; but unfortunately, this was not part of our research protocol.
Therefore, the data presented by the authors in such a small sample, in a single population, does not allow to draw any conclusion on the n-3 PUFA supplementation effect to bone mass and mineral metabolism. To investigate this effect, an ad hoc randomized trial would be required.
We fully agree that no firm conclusions can be drawn on the effect of n-3 PUFA based on this study, and hope that this comes across clearly in our discussion, as well as in the limitations section (Discussion, page 7, lines 250-265, and Conclusions, page 8, lines 267-268).
Reviewer 3 Report
Straightforward study assessing the impact of a dietary intervention on measures of bone metabolism in patients after kidney transplant. Patients received a single dose of a commercial formulation of n-3 PUFAs three times per week while placebo controls received olive oil only. Overall, there is no apparent beneficial effect of the PUFA treatment on any measure of bone metabolism. Although construed as “negative data” these findings are useful and beneficial for human patients.
Specific Comments
On Figure 1 please report what specifically is represented on the individual box plots (e.g. median).
There is no reference to measures of the distal radius for Figure 2 in the test, although that is displayed as one of the panels. In addition, why is the significant relationship between the baseline plasma level of marine n-3 PUFA and change in lumbar spine BMD not shown?
The rationale for providing a single dose according to the indicated regimen is never provided. Is it reasonable to expect there may have been a difference if a larger dose and/or increased number of doses was used?
Why were measures of calcium homeostasis not included in this study? Or at least were not reported?
Author Response
Reviewer #3
Straightforward study assessing the impact of a dietary intervention on measures of bone metabolism in patients after kidney transplant. Patients received a single dose of a commercial formulation of n-3 PUFAs three times per week while placebo controls received olive oil only. Overall, there is no apparent beneficial effect of the PUFA treatment on any measure of bone metabolism. Although construed as “negative data” these findings are useful and beneficial for human patients.
Specific Comments
On Figure 1 please report what specifically is represented on the individual box plots (e.g. median).
Reply: Thank you for your comment. The Figure text now specifies that box plots are median (line) with interquartile (box) and full range (whiskers).
There is no reference to measures of the distal radius for Figure 2 in the test, although that is displayed as one of the panels.
Reply: Thank you; the text has been corrected to specify that distal forearm is also included in this analysis.
In addition, why is the significant relationship between the baseline plasma level of marine n-3 PUFA and change in lumbar spine BMD not shown?
Reply: The positive association between baseline n-3 PUFA and change in lumbar spine BMD could point in the direction of a threshold for the effect of n-3 PUFA in bone – which was our rationale for investigating whether a treatment effect could be found if restricting analyses to patients with low baseline levels (Supplementary table 1). As this was not the case, we opted not to further highlight this finding. The Figure is attached, and has been added as supplementary data.
The rationale for providing a single dose according to the indicated regimen is never provided. Is it reasonable to expect there may have been a difference if a larger dose and/or increased number of doses was used?
Reply: The chosen regimen consisted of 3 capsules daily with 1 gram of either n-3 PUFA or olive oil. Thus, the total daily dose of EPA+DHA was approximately 2.6g, which can be considered a moderately high dosage of marine n-3 PUFA. The choice of dose was based on the primary end-point, which was an anti-inflammatory effect on the kidney and improved kidney graft function. The consideration of a potential benefit from a higher dose was weighted against the risk of side-effects, as well as the feasibility, in this cohort consisting of kidney transplant recipients, already receiving polypharmacy with immunosuppression, and other supportive medical treatment.
Why were measures of calcium homeostasis not included in this study? Or at least were not reported?
Reply: Measures of calcium metabolism included biochemical evaluation of parathyroid hormone, calcidiol, calcium ion and phosphate levels, which are included in Table 2. No differences in these biochemical markers of mineral metabolism were found based on the intervention, neither by the intention to treat, nor by the per protocol analyses. We did not include biochemical markers of bone turnover, which would have been a nice addition, but unfortunately was not part of the study protocol.

Reviewer 4 Report
Nutrients 1272743 v1. Peer Review. 11th of June 2021.
The RCT by Jørgensen et al is investigating whether supplementation with n-3 PUFAs (2.6g/day for 44 weeks) in patients with kidney transplant could provide beneficial effects to bone mineral density (BMD). The authors found no beneficial effects of the treatment on BMD, which is interesting per se, as it will provide answers to previous hypotheses. The current study will be of interest in the clinical field of kidney transplantation and would be also interesting for meta-analyses in the future. This manuscript is well written, easy to read and easy to follow. However, small adjustments will be required before publication. These are outlined point-by-point below.
Introduction. Lines 37-57. Please insert references after punctuations. eg “gender [4]” instead of “gender, [4]”. This is also found in other sections. Please edit.
Introduction. Lines 49-52. Authors should give details about treatment regimens cited. What were the doses ? Were they mixtures of DHA and EPA ? Were they either DHA or EPA ? This could explained the differences observed between these studies and the current study.
Material and Methods. Line 62. Typographical error ? What is “2015.1” ?
Material and Methods. Lines 66-69. I do not understand the calculations provided by the authors here. If participants were given 460 mg/g EPA + 380 mg/g DHA, it gives 840mg/g of n-3 PUFAs. If patients are then given 3 capsules of 1 g each per day, readers would assume the dose to be 2.52 g. However, authors wrote (in abstract and on line 67) 2.6 g of n-3 PUFAs. Is it an approximation ? If patients did indeed receive 2.6 g of total n-3 PUFAs, then authors need to explain why there might be such a difference between calculation and actual dose.
Table 1. Line 140. Authors should give truncated percentages, instead of rounded percentages. Eg university degree 37.12% (instead of 37%). Please edit the entire Table 1.
Table 2. Line s 157-158. What are the numbers in parentheses ? Is it, similar to what is written for table 1, SD ? Please explicit.
Figure 1 is very interesting (line 164). This is just a suggestion, but maybe authors would want to cluster patients between responders and non-responders to the n-3 PUFAs intervention ? Seeing the graph, it seems that the intervention might have been successful (positive % change from baseline) in some patients, while others did not respond to the intervention (negative % change from baseline). Again, this is just a suggestion, but is there any evidence in the literature of responders vs non-responders ? Could these patients be clustered into two different groups ? Countless mathematical tests could be performed, if the authors would like to investigate this further, to segregate different groups of patients ? (principal component analysis, hierarchical cluster analysis, etc…). Just a suggestion, nothing mandatory…
In Supplementary Table 1, a small typographical error can be found in “treshold”. Do authors mean “threshold” ?
Discussion, lines 186, 189, 208 please use italics when referring to Latin expression, such as “a priori”, “in vitro” and “et al”.
Discussion, lines 201-218, authors should give more details about the interventions (dose, n-3 PUFA content, type, length of treatment, etc…) given in the studies they are referring to. Indeed, different doses/regimen/lengths could explain such heterogeneity between all of these studies.
Discussion and/or Results. Authors should include in the results/discussion section(s) a small paragraph(s) on the effectiveness of their regimen (44 weeks at 2.52 g/day of n-3 PUFAs) to increase significantly n-3 PUFAs in the blood (Table 2). The effect appears to be very strong and should be highlighted somewhere in the manuscript. In addition, did the authors based their intervention regimen on a previous protocol ? If not, this is definitely worth mentioning (eg that the current intervention significantly increases n-3 PUFAs in the blood stream).
Author Response
Reviewer #4
The RCT by Jørgensen et al is investigating whether supplementation with n-3 PUFAs (2.6g/day for 44 weeks) in patients with kidney transplant could provide beneficial effects to bone mineral density (BMD). The authors found no beneficial effects of the treatment on BMD, which is interesting per se, as it will provide answers to previous hypotheses. The current study will be of interest in the clinical field of kidney transplantation and would be also interesting for meta-analyses in the future. This manuscript is well written, easy to read and easy to follow. However, small adjustments will be required before publication. These are outlined point-by-point below.
Introduction. Lines 37-57. Please insert references after punctuations. eg “gender [4]” instead of “gender, [4]”. This is also found in other sections. Please edit
Reply: Thank you, this has been corrected.
Introduction. Lines 49-52. Authors should give details about treatment regimens cited. What were the doses ? Were they mixtures of DHA and EPA ? Were they either DHA or EPA ? This could explained the differences observed between these studies and the current study.
Reply: The studies referred to in lines 49-52 are observational studies investigating the association between measured n-3 PUFA levels and bone density, thus no dosing regimens were involved. However, in the Discussion, we do briefly discuss a meta-analysis of previous interventional trials, with dosing regimens specified. Particularly, we would like to refer to the study by Chen et al, demonstrating that high dose n-3 PUFA supplementation over a two-year time period did not have any effect on bone density.
Material and Methods. Line 62. Typographical error ? What is “2015.1” ?
Reply: This was a typographical error and has been corrected, thank you.
Material and Methods. Lines 66-69. I do not understand the calculations provided by the authors here. If participants were given 460 mg/g EPA + 380 mg/g DHA, it gives 840mg/g of n-3 PUFAs. If patients are then given 3 capsules of 1 g each per day, readers would assume the dose to be 2.52 g. However, authors wrote (in abstract and on line 67) 2.6 g of n-3 PUFAs. Is it an approximation ? If patients did indeed receive 2.6 g of total n-3 PUFAs, then authors need to explain why there might be such a difference between calculation and actual dose.
Reply: The content of EPA + DHA in three capsules of 1 gram each is given as approximately 2.6 gr according to the information given by the manufacturer. We do acknowledge that this does not add up when considering the specified content of EPA and DHA. However, although there might be some variation in the measured content, we do not consider the difference between 2.52 and 2.6 g important for the effect of the intervention.
Table 1. Line 140. Authors should give truncated percentages, instead of rounded percentages. Eg university degree 37.12% (instead of 37%). Please edit the entire Table 1.
Reply: We do prefer to round percentages, as we find it improves readability. However, we have edited Table 1 to consistently included both n and (%), and hope that this increases transparency.
Table 2. Line s 157-158. What are the numbers in parentheses ? Is it, similar to what is written for table 1, SD ? Please explicit.
Reply: The numbers in parentheses are Spearman’s correlation coefficients; this is now specified in the text for clarity. In addition, we have edited the Results section throughout, including Tables, to express all data as mean±SD, median[IQI], or n (%), for consistency.
Figure 1 is very interesting (line 164). This is just a suggestion, but maybe authors would want to cluster patients between responders and non-responders to the n-3 PUFAs intervention ? Seeing the graph, it seems that the intervention might have been successful (positive % change from baseline) in some patients, while others did not respond to the intervention (negative % change from baseline). Again, this is just a suggestion, but is there any evidence in the literature of responders vs non-responders ? Could these patients be clustered into two different groups ? Countless mathematical tests could be performed, if the authors would like to investigate this further, to segregate different groups of patients ? (principal component analysis, hierarchical cluster analysis, etc…). Just a suggestion, nothing mandatory…
Reply: Thank you for these suggestions. As demonstrated in Figure 1, there is large inter-individual variability in the evolution of BMD in the first year post-transplant. This variability is not fully explained, but variations in steroid dose (Evenepoel KidneyInt 2017-91(2)-469) and the resolution of secondary hyperparathyroidism (Iyer J Am Soc Nephrol 2014-25-1331) are likely major contributors. Indeed, considering the many physiological changes occurring in the post-transplant period, any modest effect of n-3 PUFA on bone metabolism may have been particularly difficult to detect. As for additional analyses, we did perform some exploratory statistics, to investigate whether a threshold effect may be present (supplementary table 1), or whether the change in n-3 PUFA was related to the change in BMD (Figure 2). This was not the case, and to conclude, we do find it more likely that the cohort size, the limited duration of the study, and/or a modest effect size of n-3 PUFA on bone metabolism account for the lack of a positive finding.
In Supplementary Table 1, a small typographical error can be found in “treshold”. Do authors mean “threshold” ?
Reply: Thank you, corrected.
Discussion, lines 186, 189, 208 please use italics when referring to Latin expression, such as “a priori”, “in vitro” and “et al”.
Reply: Corrected throughout.
Discussion, lines 201-218, authors should give more details about the interventions (dose, n-3 PUFA content, type, length of treatment, etc…) given in the studies they are referring to. Indeed, different doses/regimen/lengths could explain such heterogeneity between all of these studies.
Reply: We completely agree that the variable dosing regimens, as well as different choices of placebo, are major limitations when trying to aggregate the rather few studies available on this subject. The paragraph mentioned here concerns experimental studies, included to inform on possible mechanisms behind an effect of n-3 PUFA on bone health. For interventional trials, we refer to the recent meta-analysis by Abdelhamid et al, and detail the dosing regimen of the only trial included in this analysis considered at low risk of bias.
Discussion and/or Results. Authors should include in the results/discussion section(s) a small paragraph(s) on the effectiveness of their regimen (44 weeks at 2.52 g/day of n-3 PUFAs) to increase significantly n-3 PUFAs in the blood (Table 2). The effect appears to be very strong and should be highlighted somewhere in the manuscript. In addition, did the authors based their intervention regimen on a previous protocol ? If not, this is definitely worth mentioning (eg that the current intervention significantly increases n-3 PUFAs in the blood stream).
Reply: Thank you for this consideration; we agree. The dose was chosen to achieve an anti-inflammatory effect, as the primary end-point was preservation of kidney graft function, taking into account the risk of side-effects, as well as the high pill-burden in this particular patient population. We have added a specification of the increase in plasma n-3 PUFA content to the results section, and specified this in the argument concerning fish-intake in the Limitations-section (Discussion, page 7, lines 216-217, and lines 259-260).
Round 2
Reviewer 1 Report
As I actually did foresee, the authors were unable to overcome my fundamental points. Neither validity nor relevance of the data presented could be substantiated.